# Use of High-Frequency Transducers in Breast Sonography

**DOI:** 10.3390/jpm12121960

**Published:** 2022-11-27

**Authors:** Antonio Corvino, Carlo Varelli, Fabio Catalano, Giulio Cocco, Andrea Delli Pizzi, Andrea Boccatonda, Fabio Corvino, Luigi Basile, Orlando Catalano

**Affiliations:** 1Movement Sciences and Wellbeing Department, University of Naples “Parthenope”, Via Medina 40, I-80133 Naples, Italy; 2Radiology Unit, Varelli Diagnostic Institute, I-80126 Naples, Italy; 3Unit of Ultrasound in Internal Medicine, Department of Medicine and Science of Aging, University “G. d’Annunzio”, I-66100 Chieti, Italy; 4Department of Innovative Technologies in Medicine and Dentistry, University “G. d’Annunzio”, I-66100 Chieti, Italy; 5Internal Medicine, Bentivoglio Hospital, AUSL Bologna, I-40010 Bologna, Italy; 6Vascular and Interventional Radiology Department, Cardarelli Hospital, I-80131 Naples, Italy; 7Advanced Biomedical Sciences Department, University Federico II of Naples, I-80138 Naples, Italy

**Keywords:** sonography, high-frequency sonography, high-resolution sonography, transducers, breast

## Abstract

Sonography companies have recently developed high-frequency transducers (20–30 MHz) to image the skin and small joints. In this pictorial review, we present a number of settings where these probes can be usefully employed to scan the breast. These include skin abnormalities of the breast and axilla; nipple–areolar complex abnormalities; superficial breast parenchyma abnormalities; breast parenchyma abnormalities in subjects with implants; very small female breasts; peripheral areas in breasts of any size; pre-puberal breasts; male breasts; post-mastectomy chest wall; and intraoperative breast sonography. Comparatively, side-by-side images obtained with conventional breast frequencies and high frequencies are shown.

## 1. Introduction

In the early years of real-time breast sonography, it was rapidly understood that high-frequency transducers were necessary to achieve an adequate axial and lateral resolution and appropriately image mammary abnormalities. A 7.5 MHz was commonly recommended in classical textbooks. Later on, with the development of digital, broad-band transducers, higher frequencies were adopted.

In a small series, Schnarkowski and coworkers [1] found a 13 MHz array to be better than a 7.5 MHz array in the differentiation and size of breast lesions. In the large, prospective study by Schulz-Wendtland et al. [2], sonography imaging at 7.5 MHz reached a sensitivity of 83% and an NPV of 84%, compared with a sensitivity of 87% and an NPV of 87% at 13 MHz. Sensitivity was especially improved in the case of small, invasive cancers (pT1a), with 78% versus 56%, respectively. Regarding tumor size, sonography at 13 MHz was also proven to have a higher accuracy in that study in comparison with sonography at 7.5 MHz [2]. The same concept applies to Doppler scanning, where it is well-known that high transmission frequencies allow the improvement of the detection of superficial, slow flows. In the study by Rjosk-Dendorfer and co-workers, the use of color Doppler breast sonography enabled cysts to be distinguished from solid masses at 17 MHz compared with 12.5 MHz, although without an improvement in differentiating benign ones from malignant ones [3].

Frequencies from 10 MHz to 15 MHz are now routinely adopted in breast sonography, allowing superficial to deep planes of breasts and their abnormalities to be displayed with a sufficiently high spatial resolution, including axial and lateral resolution, and tissue contrast, improving the differentiation of subtle shades of gray, margin resolution, and lesion conspicuity in the background of normal breast tissue [4,5,6]. However, the cost of such a high insonating frequency is decreased penetration due to the attenuation of the ultrasound (US) beam. In general, higher-frequency transducers have better spatial resolution at the cost of poor depth penetration, and lower-frequency transducers have the advantage of better depth penetration with poor spatial resolution [4]. With proper positioning and the patient in the supine or supine oblique position, most breasts are only a few centimeters thick, and high-frequency transducers provide optimum image quality for all of the breast tissue. When evaluating deep tissue in patients with particularly large breasts, it may be helpful to have lower-frequency transducers available for use only in this specific situation [5,6]. Therefore, the suggested frequency in breast imaging by the American College of Radiology practice guidelines is a center frequency of 12 MHz, precisely because of concerns regarding poor resolution at lower frequencies [7].

In the last decade, very-high-frequency probes (20 to 100 MHz) have been made commercially available and have been approved to be used in humans for optimally imaging the morphology and vascularization of very superficial structures, such as the skin, small joints, superficial tendons, and ligaments [8,9]. International guidelines in dermatology sonography highlight the need for a transmission frequency above at least 15 MHz [10,11]. However, although these probes may offer exquisite images in terms of axial and lateral resolution, their penetration capability is significantly limited (even 3 mm at 75 MHz and 1 mm at 100 MHz). This means that the full exploration of the deeper structures is not possible [9]. The use of these systems is, therefore, mostly limited to specialized senology and research units.

Since high-frequency probes have become available in our Institution, we have started to employ these frequencies under a number of circumstances when performing breast sonography.

## 2. Sonography Imaging with High-Frequency Probes

Patients were referred or self-referred to our imaging center for screening or for clinical breast sonography. Two scanners were employed, the Aplio i800 system (Canon Medical Systems Corporation, Ōtawara, Japan) and the RS85 Prestige system (Samsung Medison Co., Ltd., Seoul, Republic of Korea). The sonography examination always included a bilateral exploration of the whole breast and axilla, performed by an expert radiologist. Radial and anti-radial scans were carefully obtained, with the adjunct of the “random” exploration of the mammary and axillary area. Longitudinal and transverse scans of any abnormality identified were taken, measurements were traced, and the images were stored. Images included B-mode scans, as well as scans taken with power-Doppler or with the recently developed microvasculature-imaging systems, namely Superb Microvascular Imaging (SMI, Canon) and Microvascular Flow (MV-Flow, Samsung). The scanner was always set appropriately to detect slow flows (small color box, low pulse repetition frequency, low or null wall filter, and high color gain). Moreover, the operator added some targeted scans of the abnormalities for detection with a high-frequency transducer. For both scanners employed, this consisted of a hockey stick-shaped 22 MHz probe (i22LH8, Canon, and LA3-22AI, Samsung). The beam focus and scanning depth were modified to target the uppermost centimeters within the field of view. A large amount of gel was employed in the case of skin abnormalities without ever using any stand-off pad. Again, B-Mode and vascular-mode scans were obtained on the longitudinal and transverse plane, and the images were recorded.

## 3. Clinical Scenarios

During our breast sonography practice, we have encountered a number of settings where the adjunct of a high-frequency could be helpful, allowing improved anatomical detail.

### 3.1. Skin abnormalities of the Breast and Axilla

Any skin abnormality seen or palpated by the patient or by her family doctor becomes near-automatically alarming if it is located on the breasts or even in the axillary region. Patients become very anxious about any change seen over their breast, so, frequently, sonography exams are carried out. Sonography allows the identification, characterization, and staging of many epidermal, dermal, and hypodermal abnormalities (Figure 1 and Figure 2) [9,10,11]. These include skin tumors, sarcomas, melanoma metastases, cutaneous breast cancer recurrence, lipomas, neurofibromas, hemangiomas, foreign body reaction (piercing included), epidermal inclusion cysts, abscess, inflammation from ingrown hair, and suppurative hidradenitis.

With rare exceptions, dermal lesions are benign. For lesions that appear to be located within the subcutaneous layer, evaluating the angles that are made between the lesion margin and the dermis is helpful; typically, lesions that are completely within the hypodermis form an obtuse angle with the dermis as they push upward, whereas lesions that are partially within the dermis and partially within the subcutaneous fat form an acute angle with the dermis [12]. Additionally, a “claw” of dermal tissue that wraps around the margin of the lesion is suggestive of a dermal origin. Epidermal inclusion cysts are located in the subcutaneous region as well, but, particularly if imaged with a high-frequency probe, they show a tract reaching the epidermal surface [12,13].

Sonography has been employed to investigate dermal thickening in the case of post-surgical lymphedema at 18 MHz [14] and acute radiation dermatitis at 18 MHz and 20 MHz (Figure 3) [15].

### 3.2. Nipple–Areolar Complex Abnormalities

Nipple tumors are uncommon. They include adenoma, hemangioma, and melanoma. The retroareolar region is not easy to scan with sonography because of the image deformation and shadowing created by the nipple. Compressing the para-areolar skin and angling the probe toward the retroareolar region can be a valuable tip, as well as gently pulling the nipple with the hand not handling the probe [14]. Using a higher-frequency probe may allow the exploration of the retroareolar region to be improved (Figure 4) [16]. 

Ductal ectasia is a frequently encountered abnormality (Figure 5). Intraductal lesions include solidified secretion, ductal hyperplasia, intraductal papilloma, and carcinoma (both in situ and invasive) (Figure 6) [17,18]. The specificity of sonography in differentiating these abnormalities is low (33%), as most lesions are small and have overlapping features [17]. Malignant lesions are usually larger, with non-circumscribed margins, mixed hyperechoic–hypoechoic or complex echogenicity, and significant vascularity [17]. Recognizing intralesional flow signals allows a solidified secretion to be ruled out. On the other hand, the more the vascularization is detected, the more the suspicion of malignancy will be, prompting biopsy. Being very sensitive to slow flows, SMI has proven effective in the characterization of intraductal lesions. SMI with the Vascular Index tool, which allows the quantification of the number of colored pixels within the color box, yielded a sensitivity of 71% and a specificity of 49% in the prediction of malignancy in intraductal lesions [18].

### 3.3. Superficial Breast Parenchyma Abnormalities

Breast parenchyma lesions can be located at variable depths. When particularly superficial, these abnormalities can be scanned with a better resolution using a high-frequency probe (Figure 7 and Figure 8). When the growth epicenter of a lesion is in the echogenic dermal layer, it cannot be due to breast cancer. Instead, when a lesion is in a subcutaneous location, it cannot be ruled out as a breast tumor. In fact, the uppermost duct lobular units can expand into the hypodermis along Cooper ligaments or may persist adjacent to these connective bands when the breast parenchyma undergoes post-menopausal involution. Consequently, a careful sonography assessment of all non-dermis abnormalities is always mandatory [12,13]. 

### 3.4. Breast Abnormalities in Subjects with Implants

Implants are frequently placed surgically below the gland or the muscular plane to augment the breast volume for aesthetic reasons. Additionally, subjects submitted to radical breast surgery for cancer frequently undergo immediate or two-stage reconstruction through the placement of a submuscular prosthesis. In the latter circumstance, there is no breast parenchyma anymore, while after breast augmentation surgery, the parenchyma, usually thin by itself, is displaced superficially and compressed to a variable degree by the implant. In both cases, any abnormality will be, in the end, quite superficial and it can be optimally scanned using high-frequency probes (Figure 9 and Figure 10). Demonstrating flow signals inside a breast abnormality allows a dirty cyst to be identified, an active, growing fibroadenoma to be predicted, or a malignant lesion to be suspected, especially in the case of marked and anarchic vascular architectures. Particularly after demolitive breast cancer surgery, local recurrence can develop, requiring a prompt diagnosis (Figure 11). 

### 3.5. Small-Size Breasts

In the supine or oblique patient position, the thickness of most breast tissue does not exceed 3–5 cm [5]. Not uncommonly, however, including both in reproductive-age and post-menopausal women, the amount of tissue between the cutis–subcutis layer and the chest wall is very thin. Consequently, the breast and its abnormalities can be appropriately scanned using higher-frequency probes. Even in larger breasts, the peripheral area is quite thin and can be usefully imaged with high-frequency transducers.

The same concept applies to pre-puberal breasts. Before puberal enlargement of the gland, the breast is thin and any abnormality can be optimally evaluated using high frequencies. Anatomic variations, premature telarca, puberal gynecomastia, retroareolar Montgomery cysts, ductectasia, retroareolar abscess, hemangioma, and fibroadenoma can be adequately investigated.

Finally, the male breast is also thin, except in the case of marked lipomastia or gynecomastia. Male breast abnormalities are usually located in the retroareolar region, but they can develop in any area. In clinical practice, the main need is to differentiate unilateral, or apparently unilateral, gynecomastia from breast carcinoma. Other uncommon abnormalities of the male breast include abscess, hematoma, diabetic mastopathy, foreign bodies, and benign tumors. All these changes are optimally depicted using high-frequency probes (Figure 12) [4].

### 3.6. Post-Mastectomy Chest Wall

After radical mastectomy, the chest wall is routinely scanned to detect a local recurrence or to evaluate any other palpable or non-palpable abnormality. In postmastectomy patients, any irregular mass, vascularity, and occurrence involving chest wall layers suggest malignancy and can be more accurately assessed using higher-frequency probes [19]. 

### 3.7. Intra-Operative Breast Sonography

Intra-operative sonography is an effective technique for localizing non-palpable breast masses, facilitating excision and increasing radicality by assessing the resection margins [20]. Given the very short distance between the probe and the area of interest, intra-operative sonography required the use of high-frequency transducers.

## 4. Disadvantages

High-frequency probes bear some drawbacks. We do not encourage the use of these transducers as the only tool to explore small breasts or male breasts. Their footprint is small, consisting of 8 mm, so a whole-breast scanning would be time-consuming and at risk of missing some abnormality. Breast exploration should always be carried out with conventional sonography probes, with the adjunct of a high-frequency system only in the case of superficial abnormalities when, on the basis of their expertise, the operator feels that further information can be obtained also using high frequencies. Moreover, it should be considered that these probes are more sensitive to moving artifacts. Particularly when assessing the vascularization of a given nodule or area, care must be taken to avoid the movement of the operator’s hand or the transducer. Additionally, the probe must always be placed gently over the skin’s surface to avoid the distortion of the US beam and compression of the flow signals inside thin vessels. Finally, and obviously, the higher the frequency of the probe is, the lower the beam penetration will be; consequently, using a high-frequency transducer in the case of intermediate to deep breast abnormalities will not be of any additional value. 

## 5. Conclusions

If a high-frequency (20–30 MHz) probe is available, it should be used in a targeted way to image breasts. Very fine scans can be obtained, with optimal spatial resolution and anatomic detail. Prospective studies are needed to establish the additional practical value of high-frequency probes in breast imaging and to assess their impact on the BI-RADS categorization of the lesions. 

## Figures and Tables

**Figure 1 jpm-12-01960-f001:**
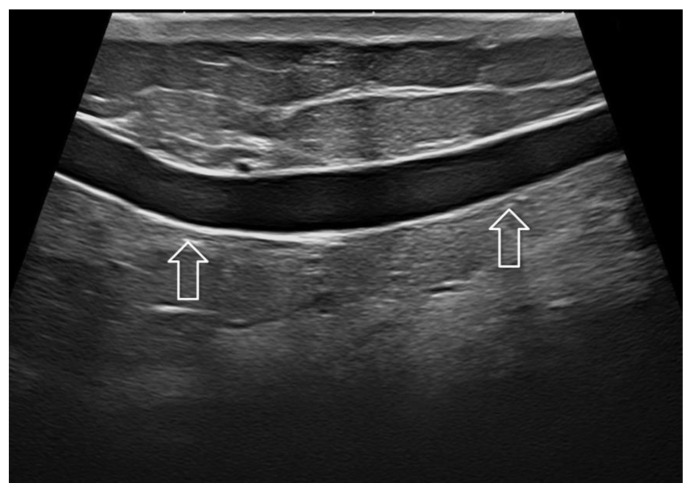
Breast vein imaged at 22 MHz. Note the high-quality details of the venous wall and lumen.

**Figure 2 jpm-12-01960-f002:**
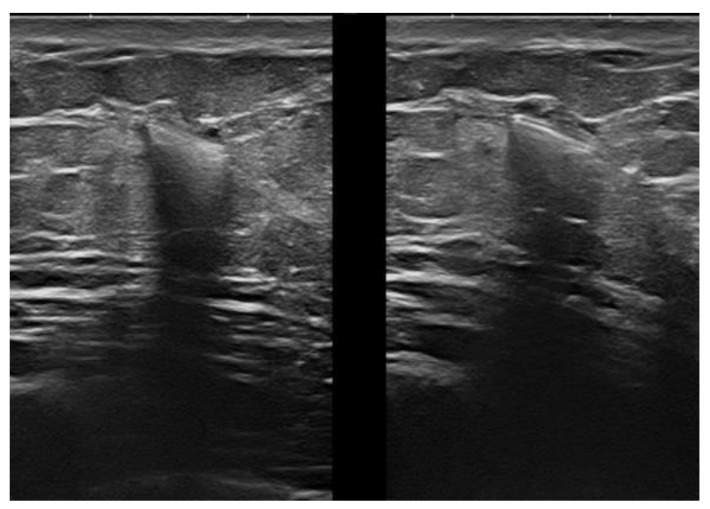
Surgical resection for breast cancer, split screen mode. Detailed depiction of the scarring, with the optimal display of a surgical clip causing reverberation artifacts.

**Figure 3 jpm-12-01960-f003:**
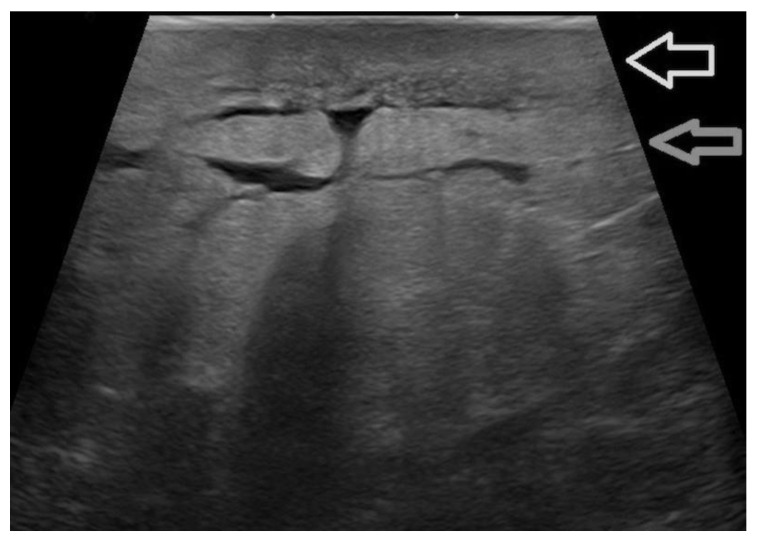
Skin thickening after conservative breast surgery and adjuvant irradiation. Imaging at 22 MHz allowed an optimal depiction of the thickened dermis (white arrow) and hypodermis (gray arrow).

**Figure 4 jpm-12-01960-f004:**
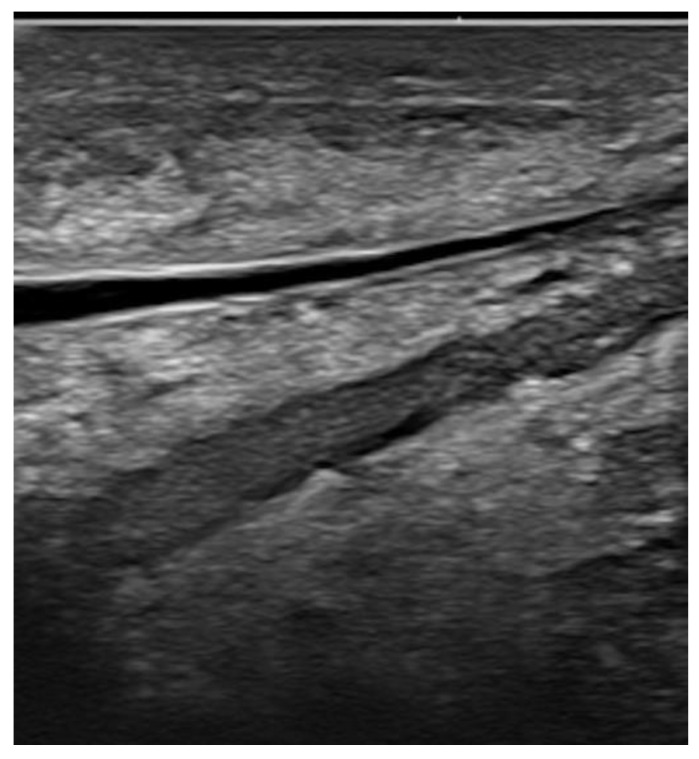
Breast ducts. The uppermost shows an anechoic content, while the other is filled with amorphous material.

**Figure 5 jpm-12-01960-f005:**
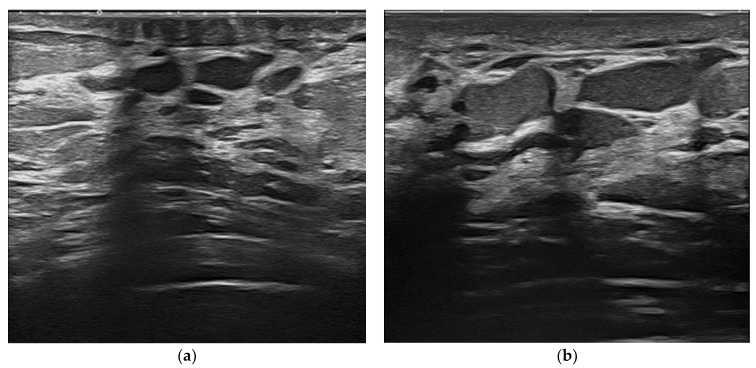
Ductectasia. Dilated ducts filled with dense, fluid material as they appear at 15 MHz (**a**) and at 22 MHz (**b**).

**Figure 6 jpm-12-01960-f006:**
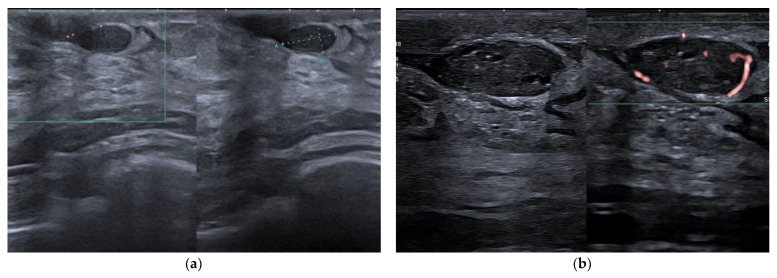
Intraductal papilloma. Retroareolar, oval hypoechoic nodule without flow signals (8 × 4 mm). Appearance at 15 MHz (**a**). When imaged at 22 MHz (**b**), the lesion is clearly located in the lumen of a duct and exhibits some internal vessels under power-Doppler imaging.

**Figure 7 jpm-12-01960-f007:**
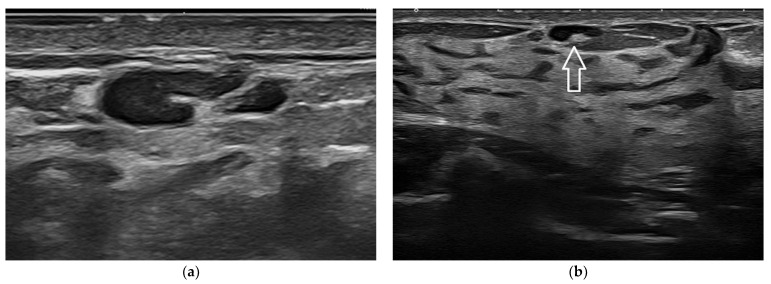
Normal intramammary lymph node. Being very superficial, this lymph node is better depicted in its cortex and hilum anatomy when imaged at 22 MHz (**a**) than at 15 MHz (**b**).

**Figure 8 jpm-12-01960-f008:**
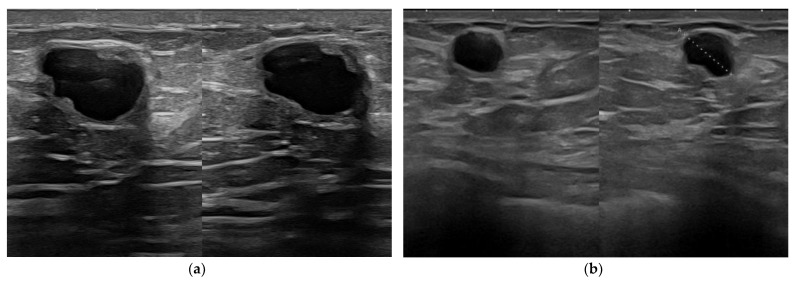
Complex fluid lesion due to post-surgical steatonecrosis. Given the superficial location of the abnormality, it is better depicted in its echotexture when imaged at 22 MHz (**a**) than at 15 MHz (**b**).

**Figure 9 jpm-12-01960-f009:**
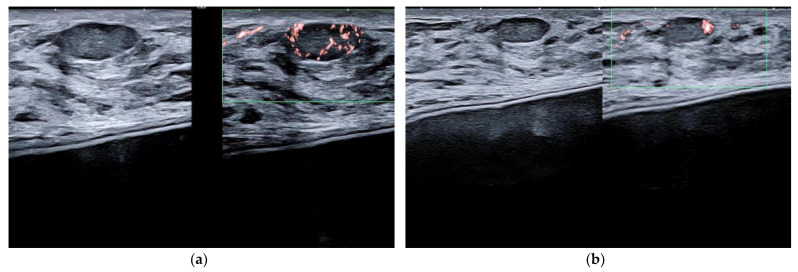
Fibroadenoma in a patient with a dual-plane breast implant. The quite superficial nodule is better depicted in its echotexture and vascularization by power-Doppler imaging when imaged at 22 MHz (**a**) than at 15 MHz (**b**).

**Figure 10 jpm-12-01960-f010:**
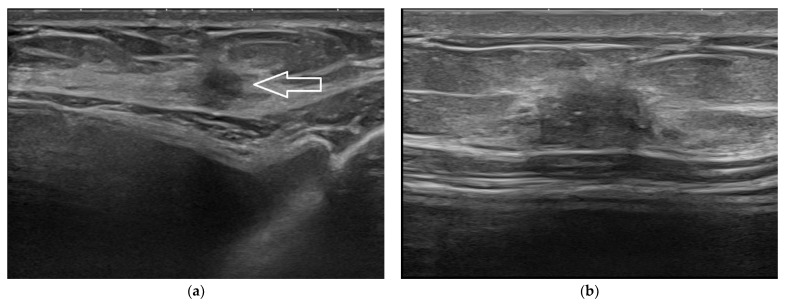
An irregular distortion detected in a patient with a history of breast augmentation surgery and a submuscular implant (**a**, arrow). The lesion is better displayed at 22 MHz (**b**), where the spiculated margins, the hyperechoic desmoplastic border, and the internal echoic spots due to microcalcification are better appreciated. Histology confirmed an invasive ductal carcinoma.

**Figure 11 jpm-12-01960-f011:**
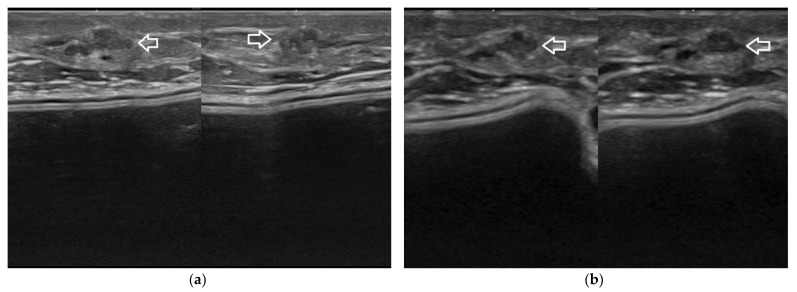
Local cancer recurrence in a patient with a history of subcutaneous mastectomy and subsequent breast implantation reconstruction. The tiny tumor foci are better depicted at 22 MHz (**a**) than at 15 MHz (**b**, arrows).

**Figure 12 jpm-12-01960-f012:**
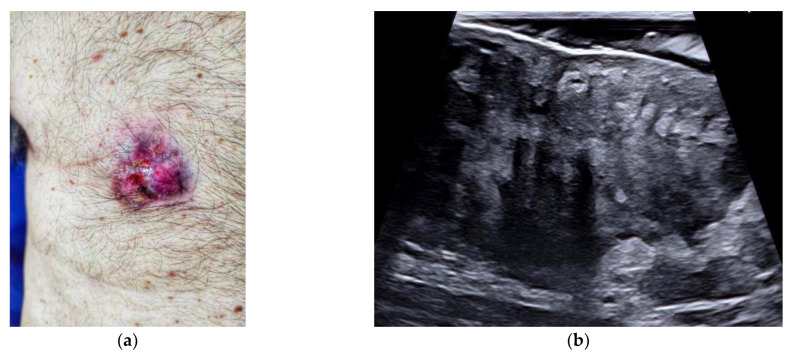
Locally advanced male breast cancer. Clinical picture (**a**), high-resolution sonography scan (**b**), and computed tomography scan in the portal phase of the breast lesion (**c**). The use of the high-frequency transducer allows the depiction of the tumor tissue infiltrating the nipple–areolar complex.

## Data Availability

Not applicable.

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
