# Peer review of "Use of High-Frequency Transducers in Breast Sonography"

_jpm, 2022, doi:10.3390/jpm12121960_

Round 1

Reviewer 1 Report (New Reviewer)

The authors have presented high-frequency ultrasound imaging advantage in finding multiple biomarkers in breast sonography with adequate Figures. The authors have shown that high-resolution ultrasound imaging is helpful in the differentiation of several anatomical features. This work will surely be informative for the ultrasound community. I recommend it for publication; however, I have the following minor comments.   

  1. It is not clear what the authors mean in lines 42 & 43
  2. Authors should comment on the difference between the left and right subfigures of Figure 2.

Author Response

Dear Editor, dear Reviewers,

we are pleased that you are willing to reconsider our manuscript for publication, pending MAJOR revisions. After giving careful consideration to the points that the referees have made, we revised our article according to reviewer’ recommendations.

We hope very much that you and your colleagues will find our changes adequate. If you deem it necessary, we are willing to make further additions.

REVIEWER 1

The authors have presented high-frequency ultrasound imaging advantage in finding multiple biomarkers in breast sonography with adequate Figures. The authors have shown that high-resolution ultrasound imaging is helpful in the differentiation of several anatomical features. This work will surely be informative for the ultrasound community. I recommend it for publication; however, I have the following minor comments.  

1)It is not clear what the authors mean in lines 42 & 43

2)Authors should comment on the difference between the left and right subfigures of Figure 2.

MANUSCRIPT REVISION

REVIEWER #1

1)It is not clear what the authors mean in lines 42 & 43

[rev1, comment1]

According to your suggestion, we made the sentence clearer. Now the text is the following: “In the study by Rjosk-Dendorfer and co-workers, the use of color Doppler breast sonography enabled to distinguish cysts from solid masses at 17 MHz compared to 12.5 MHz, although without an improvement in differentiating benign from malignant ones”.

___________________________________________________________________

2)Authors should comment on the difference between the left and right subfigures of Figure 2.

[rev1, comment2]

Left and right are two separate figures of the same scar and surgical clip. We specified in the text that split screen mode was used.

Reviewer 2 Report (New Reviewer)

see attached file

Author Response

Dear Editor, dear Reviewers,

we are pleased that you are willing to reconsider our manuscript for publication, pending MAJOR revisions. After giving careful consideration to the points that the referees have made, we revised our article according to reviewer’ recommendations.

We hope very much that you and your colleagues will find our changes adequate. If you deem it necessary, we are willing to make further additions.

REVIEWER 2

Overview

The authors present a pictorial review of the use of high-frequency (20-30 MHz) ultrasound for in-vivo breast imaging. Multiple examples of images achieved with this method are presented. There is discussion of the situations in which this type of imaging would be useful.

As the availability of this frequency range is relatively new, and clinical use is rare, these results and discussion are interesting, especially when compared with results using more conventional (lower) frequencies. However, as outlined in my review below, I would expect more thoroughness from the text part of this review article, and I would like to see improvements to the images presented (as outlined in my review below) before considering publication of the manuscript. I also include minor comments regarding the clarity and accuracy. In general, the authors might want to consider having a native english speaker review the manuscript for grammatical errors.

Main questions/comments

(1) Introduction

a)The authors mention that frequencies in ultrasonography have increased with time, and that groups have found that higher-frequency arrays were ‘better’ than lower- frequency ones, but they do not say why this is (higher frequencies allow for better spatial resolution). Although basic, this is a key point that needs to be included.

b)This is also a good place to talk about the disadvantage of higher frequency (lower penetration), as that point is clear from some of the images. Later (line 198) this concept is also referred to without explanation. Only in the very last sentence is this mentioned. I would like to see a thorough review of what resolution we can expect from these transducers, with some of the basic physics being reviewed, as is often done in the intro sections of reviews on ultrasound imaging. What kinds of structures might we expect to see with this that can’t be seen by lower frequencies? I know that we see some examples of these in the subsequent images, but this should be summarized in the intro.

c)I’m not sure what point is being made by mentioning the Schulz-Wendtland study, as the sensitivity was indeed (slightly) higher for higher frequencies, but the specificity decreased, so those seem like conflicting results. There is also no citation for that study.

d)Line 43: without an improvement at higher frequencies? Doesn’t this then contradict the point being made?

e)Line 45: “all companies” is vague, and also not backed up by citations or other proof.

f)Line 45: “frequency allowable” doesn’t mean anything. Is this the lowest frequency? The recommended one?

g)Line 48-9: “most of manufacturers” and “basically 20 MHz to 30 MHz” are both too vague.

(2) Images

a)Because the main aspect of this “pictoral review” is the images, can they be larger so we can see some of the details referred to?

b)In e.g. Figure 1, it would be preferable to highlight areas of interest with arrows or coloured boxes. Failing that, if a spatial scale was included (x and z), the caption could tell us where these areas are in terms of (x,z).

c)I actually think that all figures should have a spatial scale, in mm. This is because a main interest is the identification and resolution of spatial structures in the breast.

d)Figure 2: are these two separate figures of the same thing?

e)For figures which compare two frequency ranges (e.g. Fig. 5 & 6), the whole point is to be able to compare the images, but they are shown at very different spatial scales. I appreciate that the authors want to show the different depths available from the different frequencies. But what is needed is to show a zoom of the low-frequency image to a region which approximately matches that of the high-frequency image, so that the advantage of the high-frequency can be seen by eye.

f)Red regions (I presume from Doppler imaging?) are unexplained

g)Figure 11 is mostly black space, and then the details we’re asked to look at are very tiny.

h)Figure 12: what is the arrow, and how does this image relate to the ultrasound one, or help us understand it?

(3) Minor questions/comments

(1)  Line 152 “it should be considered that”–by whom? Why should this be considered?

(2)  Line 161: consider defining BI-RADS, as that’s the first time it shows up in the article.

(3)  Line 76: what are “appropriately slow flows”? And “whenever felt useful”?

(4)  Line 94: why would a determatology visit fit better? And what is the point of this sentence when the next one seems to contradict it?

(5)  Line 174: “any abnormality will be in the end quite superficial” is much too vague.

(6)  Line 211 and 251: “optimal” is an overclaim unless it can be backed up: was the resolution only diffraction-limited, and contrast ideal?

(7)  Line 220-224: what is the point of this subsection?

(8)  Line 220 needs a citation

(9) Line 260: why are they more sensitive, and how bad is this?

(10) Line 243: not sure what is meant by “gray-scale finding”.

The following do not require responses from the authors:

(1) There are many commas in places where they do not need to be – this is especially noticeable in the introduction.

(2) Writing “allows to”or“enables to” is incorrect.

MANUSCRIPT REVISION

REVIEWER #2

Main questions/comments

(1) Introduction

a)The authors mention that frequencies in ultrasonography have increased with time, and that groups have found that higher-frequency arrays were ‘better’ than lower- frequency ones, but they do not say why this is (higher frequencies allow for better spatial resolution). Although basic, this is a key point that needs to be included.

[rev2, comment1a]

You are right. The introductory paragraph was confusing. We detailed the resolution-depth trade-off by adding the following sentence:

“However, the cost of such a high insonating frequency is decreased penetration due to attenuation of the ultrasound beam. In general, higher-frequency transducers have better spatial resolution at the cost of poor depth penetration, and lower-frequency transducers have the advantage of better depth penetration with poor spatial resolution”.

Indeed, the whole introductory paragraph has been rewritten with an in-depth focus on the concepts of spatial resolution and contrast obtained through the use of high and very high frequency probes. We hope that the changes can be considered adequate. If you deem it necessary, we are willing to make further additions.

___________________________________________________________________

b)This is also a good place to talk about the disadvantage of higher frequency (lower penetration), as that point is clear from some of the images. Later (line 198) this concept is also referred to without explanation. Only in the very last sentence is this mentioned. I would like to see a thorough review of what resolution we can expect from these transducers, with some of the basic physics being reviewed, as is often done in the intro sections of reviews on ultrasound imaging. What kinds of structures might we expect to see with this that can’t be seen by lower frequencies? I know that we see some examples of these in the subsequent images, but this should be summarized in the intro.

[rev2, comment1b]

The whole introductory paragraph has been rewritten. The disadvantage of higher frequency has been addressed. Now, the text is the following:

“Frequencies from 10 MHz to 15 MHz are now routinely adopted in breast sonography, allow displaying from superficial to deep planes of breast and their abnormalities with a sufficiently high spatial resolution, including axial and lateral resolution, and tissue contrast, permitting improved differentiation of subtle shades of gray, margin resolution, and lesion conspicuity in the background of normal breast tissue. However, the cost of such a high insonating frequency is decreased penetration due to attenuation of the ultrasound beam. In general, higher-frequency transducers have better spatial resolution at the cost of poor depth penetration, and lower-frequency transducers have the advantage of better depth penetration with poor spatial resolution. With proper positioning and the patient in the supine or supine oblique position, most breasts are only a few centimeters thick and high-frequency transducers provide optimum image quality for all the breast tissue. When evaluating deep tissue in patients with particularly large breasts, it may be helpful to have lower frequency transducers available to be used only in this specific situation. Therefore, the suggested frequency in breast imaging by the American College of Radiology practice guidelines is a center frequency of 12 MHz precisely because of concerns regarding poor resolution at lower frequencies.

In the last decade, very high-frequency probes (20 to 100 MHz) have been made commercially available and are approved to be used in humans, to optimally image the morphology and vascularization of very superficial structures such the skin, small joints, superficial tendons and ligaments.8,9 International guidelines in dermatology sonography highlight the need for a transmission frequency at least above 15 MHz. However, although these probes may offer exquisite images in terms of axial and lateral resolution, their penetration capability is significantly limited (even 3 mm at 75 MHz and 1 mm at 100 MHz). This means that the full exploration of the deeper structures is not possible. The use of these systems is, therefore, mostly limited to specialized senology and research units”.

We hope that the changes can be considered adequate. If you deem it necessary, we are willing to make further additions.

___________________________________________________________________

c)I’m not sure what point is being made by mentioning the Schulz-Wendtland study, as the sensitivity was indeed (slightly) higher for higher frequencies, but the specificity decreased, so those seem like conflicting results. There is also no citation for that study.

[rev2, comment1c] 

You are right. The phrase was misleading. Now, we reported it as it is discussed in the original article, underlining the positive results obtained by the use of 13MHz probes in terms of sensitivity and PNV (according to the authors, almost with the same specificity and PPV than the 7,5 MHz probes). We added also the reference. Now, the text is as follows:

“In the large prospective study from Schulz-Wendtland et al. sonography imaging at 7,5 MHz reached a sensitivity of 83% and a NPV of 84%, compared to a sensitivity of 87% and a NPV of 87% at 13 MHz. Sensitivity was especially improved in case of small invasive cancers (pT1a) with 78% versus 56%, respectively.2 Regarding tumor size, sonography at 13 MHz proved to have a higher accuracy in that study in comparison with sonography at 7,5 MHz”.

___________________________________________________________________

d)Line 43: without an improvement at higher frequencies? Doesn’t this then contradict the point being made?

[rev2, comment1d]

You are right. According to your suggestion, we made the sentence clearer. Now, the text is the following:

“In the study by Rjosk-Dendorfer and co-workers, the use of color Doppler breast sonography enabled to distinguish cysts from solid masses at 17 MHz compared to 12.5 MHz, although without an improvement in differentiating benign from malignant ones”. Specifically, the authors of the article emphasized elastography rather than color Doppler, so they did not go too far on this aspect.

___________________________________________________________________

e)Line 45: “all companies” is vague, and also not backed up by citations or other proof.

[rev2, comment1e]

We removed "presets by all companies" from the sentence.

 ______________________________________________________________

f)Line 45: “frequency allowable” doesn’t mean anything. Is this the lowest frequency? The recommended one?

[rev2, comment1f]

We replaced allowable with “suggested”. I want to clarify that we used “allowable” because it says so right on the ACR document.

___________________________________________________________________

g)Line 48-9: “most of manufacturers” and “basically 20 MHz to 30 MHz” are both too vague.

[rev2, comment1g]

We replaced the sentence. Now, the text is the following:

“In the last decade, very high-frequency probes (20 to 100 MHz) have been made commercially available and are approved to be used in humans, to optimally image the morphology and vascularization of very superficial structures such the skin, small joints, superficial tendons and ligaments”.

___________________________________________________________________

(2) Images

a)Because the main aspect of this “pictoral review” is the images, can they be larger so we can see some of the details referred to?

[rev2, comment2a]

We didn't get it right. Are you referring to the possibility of increasing the resolution of images? If so, consider that all the images are already set to 300dpi. On the other hand, if you are referring to the possibility of enlarging the image, consider that the size of the images (800x800 pixel) is dictated by editorial layout needs. Too large images will decrease the resolution too much. In any case, the size of the images can be easily modified in the editorial phase by staff.

___________________________________________________________________

b)In e.g. Figure 1, it would be preferable to highlight areas of interest with arrows or coloured boxes. Failing that, if a spatial scale was included (x and z), the caption could tell us where these areas are in terms of (x,z).

[rev2, comment2b]

According to your suggestion, we inserted an arrow indicating the vein.

___________________________________________________________________

c)I actually think that all figures should have a spatial scale, in mm. This is because a main interest is the identification and resolution of spatial structures in the breast.

[rev2, comment2c]

We agree with your considerations. However, when preparing the article, we cropped the images to maximize resolution and avoid any unnecessary details being visible, for example the identity of the patients. Uncropping the scans would make loose the image quality and the education value of our pictorial review. Regarding the scale in mm, aim of this pictorial review was the representation of the usefulness of these high-resolution probes in various clinical scenarios. We think we have already gone into too much detail in the introductory part by entering into medical engineering concepts (axial and lateral resolution, contrast resolution, trade off resolution-depth), for which we had to consult the specialist of the equipment in our endowment.

___________________________________________________________________

d)Figure 2: are these two separate figures of the same thing?

[rev2, comment2d]

Left and right are two separate figures of the same scar and surgical clip. We specified in the text that split screen mode was used.

______________________________________________________________

e)For figures which compare two frequency ranges (e.g. Fig. 5 & 6), the whole point is to be able to compare the images, but they are shown at very different spatial scales. I appreciate that the authors want to show the different depths available from the different frequencies. But what is needed is to show a zoom of the low-frequency image to a region which approximately matches that of the high-frequency image, so that the advantage of the high-frequency can be seen by eye.

[rev2, comment2e]

That’s a good point. However, I preferred to enlarge uniformly all the images of the paper. If I cut or zoom the images, the journal reader would not have the idea of the real advantage offered by the use of upper range of high-resolution probes since the main focus of the article is to emphasize their utility in breast imaging.

______________________________________________________________

f)Red regions (I presume from Doppler imaging?) are unexplained

[rev2, comment2f]

Thank you for the clarification. I added in corresponding figure legends the sentence “at power doppler imaging”.

__________________________________________________________________

g)Figure 11 is mostly black space, and then the details we’re asked to look at are very tiny.

[rev2, comment2g]

In figure legend I didn’t specify that the patient was subjected to a mastectomy and after to a breast implantation. I added it into the corresponding legend so to explain the posterior black space.

___________________________________________________________________

h)Figure 12: what is the arrow, and how does this image relate to the ultrasound one, or help us understand it?

[rev2, comment2h]

These three images are the different appearance of the male breast cancer seen clinically in figure a, both in US imaging and portal phase CT contrast image. I added it the following sentence: “in portal phase of the breast lesion”.

___________________________________________________________________

(3) Minor questions/comments

(1)  Line 152 “it should be considered that”–by whom? Why should this be considered?

[rev2, comment3(1)]

Thank you for the suggestion. I deleted this sentence.

______________________________________________________________

(2)  Line 161: consider defining BI-RADS, as that’s the first time it shows up in the article.

[rev2, comment3(2)]

Thank you for the suggestion; however, this is not the main point of the article. I preferred to delete it in the corresponding figure legend.

______________________________________________________________

(3)  Line 76: what are “appropriately slow flows”? And “whenever felt useful”?

[rev2, comment3(3)]

To explain the first sentence, I added to the text the following details: “(small color box, low pulse repetition frequency, low or null wall filter, high color gain)” to explain the slow flow preset setting which we used. I deleted the second sentence “whenever felt useful” replacing it with “moreover”.

______________________________________________________________

(4)  Line 94: why would a dermatology visit fit better? And what is the point of this sentence when the next one seems to contradict it?

[rev2, comment3(4)]

I deleted it; I agree with you that it makes no sense.

______________________________________________________________

(5)  Line 174: “any abnormality will be in the end quite superficial” is much too vague.

[rev2, comment3(5)]

I disagree with this sentence because in breast augmentation after mastectomy the residual parenchyma is compressed and displace more superficially; this sentence is explained in the relative paragraph “Breast abnormalities in subject with implants”.

______________________________________________________________

(6)  Line 211 and 251: “optimal” is an overclaim unless it can be backed up: was the resolution only diffraction-limited, and contrast ideal?

[rev2, comment3(6)]

I added reference 4 to validate the sentence in which there is the term “optimal” in line 211. We already detailed the resolution-depth trade-off re-writing the “Introduction” paragraph.

______________________________________________________________

(7)  Line 220-224: what is the point of this subsection?

[rev2, comment3(7)]

The main point of this subsection is to emphasize the role of US in studying the chest wall of patients underwent to mastectomy.

___________________________________________________________________

(8)  Line 220 needs a citation

[rev2, comment3(8)]

I added it, please see the rev2, comment3(6)

______________________________________________________________

(9) Line 260: why are they more sensitive, and how bad is this?

[rev2, comment3(9)] 

The high frequency US probes are very small as compared to standard one, thus they are more sensitive to motion artifact. However, this is a technical characteristic to know, but nothing to be considered a real limitation. Indeed, in the text we specified this with the following sentence: “when assessing the vascularization of a given nodule or area, care must be taken to avoid motion of the operator’s hand or the transducer”.

______________________________________________________________

(10) Line 243: not sure what is meant by “gray-scale finding”.

[rev2, comment3(10)]

We referred to the artifacts of the US beam distortion and near-field artifacts.

______________________________________________________________

Round 2

Reviewer 2 Report (New Reviewer)

I agree with all changes made to the text by the authors, who have answered most of my questions and comments (thank you). I have two comments which I think should be addressed before publication:

(1) I still don't understand the point of the section "Post-mastectomy chest wall", as the use of high-frequency transducers is not mentioned at all, and doesn't refer to any of the figures in the text. If there is no relation to using high-frequency transducers, I think this section should be taken out.

(2) The English still needs substantial correction. But if the journal editors are happy with the English quality, or are happy to do the corrections, then I do not object to publication.

(3) Yes, my comment about image size was more for the editorial staff. It would be ideal if the images were larger, but I realize that there is a specific journal format in place.

(4) Fig. 11:  in the right-hand size images (I presume this is the right breast?) the the arrows for (a) and (b) seem to point to different structures

(5) Define the acronym US (last paragraph of introduction)

Author Response

Dear Editor, dear Reviewers,

we are pleased that you are willing to reconsider our manuscript for publication, pending MINOR revisions. After giving careful consideration to the points that the referees have made, we revised our article according to reviewer’ recommendations.

We hope very much that you and your colleagues will find our changes adequate. If you deem it necessary, we are willing to make further additions.

REVIEWER 2

I agree with all changes made to the text by the authors, who have answered most of my questions and comments (thank you). I have two comments which I think should be addressed before publication:

(1) I still don't understand the point of the section "Post-mastectomy chest wall", as the use of high-frequency transducers is not mentioned at all, and doesn't refer to any of the figures in the text. If there is no relation to using high-frequency transducers, I think this section should be taken out.

(2) The English still needs substantial correction. But if the journal editors are happy with the English quality, or are happy to do the corrections, then I do not object to publication.

(3) Yes, my comment about image size was more for the editorial staff. It would be ideal if the images were larger, but I realize that there is a specific journal format in place.

(4) Fig. 11:  in the right-hand size images (I presume this is the right breast?) the the arrows for (a) and (b) seem to point to different structures.

(5) Define the acronym US (last paragraph of introduction).

MANUSCRIPT REVISION

REVIEWER #2

(1) I still don't understand the point of the section "Post-mastectomy chest wall", as the use of high-frequency transducers is not mentioned at all, and doesn't refer to any of the figures in the text. If there is no relation to using high-frequency transducers, I think this section should be taken out.

[rev2, comment1]

You’re right. The paragraph was confusing. I preferred to edit the text rather than delete the paragraph. Now, the text is as follows: “After radical mastectomy the chest wall is routinely scanned to detect a local recurrence or to evaluate any other palpable or not palpable abnormality. In postmastectomy patients, any irregular mass, vascularity, and occurrence involving chest wall layers suggest malignancy and can be more accurately assessed using higher frequency probes”.

_____________________________________________________________________

(2) The English still needs substantial correction. But if the journal editors are happy with the English quality, or are happy to do the corrections, then I do not object to publication.

[rev2, comment2]

A linguistic review by a native English speaker was carried out.

_____________________________________________________________________

(3) Yes, my comment about image size was more for the editorial staff. It would be ideal if the images were larger, but I realize that there is a specific journal format in place.

[rev2, comment3]

We increased the size of some images, but I think it will be the one who will paginate the article to decide the final format.

_____________________________________________________________________

(4) Fig. 11:  in the right-hand size images (I presume this is the right breast?) the the arrows for (a) and (b) seem to point to different structures.

[rev2, comment4]

 No, there are two different small neoplastic foci.

_____________________________________________________________________

(5) Define the acronym US (last paragraph of introduction).

[rev2, comment5]

okay, done.

_____________________________________________________________________

This manuscript is a resubmission of an earlier submission. The following is a list of the peer review reports and author responses from that submission.

Round 1

Reviewer 1 Report

Dear authors,

Thank you for your work.

There are a few areas where grammar and sentence structure could be improved for clarity.

For example,

1.  Pg 1, line 41: In the study from...(sic)

May i suggest: In the study by...

2.  Pg 3 Line 92: Patients becomes very anxious for any change seen over theur breast (sic)...

3.  Pg 7 line 154: As a matter of facts (sic)

4.  Pg 11, line 172: ...immediate or two-times (sic)...

Perhaps it should be phrased: 'immediate or two-stage reconstruction', or ...'reconstruction in two stages'...

5.  Pg 18, line 240: ...'basing on experience'...

6. Pg 18, line 252: ...'it can be usefully be employed'...

Here again both sentence structures are not ideal.

Apart from the above minor grammatical issues, there is reference to 

7. Figure 9. It might be good for the authors to explain why detecting vascularity in what is assumed to be a fibroadenoma is considered an improvement in imaging.  Conventionally, vascularity is a characteristic of indeterminate lesions which may warrant biopsy.  

8. Figure 10.  Was the final diagnosis listed in the text or in the legend? It would be interesting to know the histology for this irregular lesion.

Thank you.

Author Response

Dear Editor, dear Reviewers,

we are pleased that you are willing to reconsider our manuscript for publication, pending MAJOR revisions. After giving careful consideration to the points that the referees have made, we revised our article according to reviewer’ recommendations.

We hope very much that you and your colleagues will find our changes adequate.

REVIEWER 1

Dear authors,

Thank you for your work.

There are a few areas where grammar and sentence structure could be improved for clarity.

For example,

  1. Pg 1, line 41: In the study from...(sic)

May i suggest: In the study by...

  1. Pg 3 Line 92: Patients becomesvery anxious for any change seen over theur breast (sic)...
  2. Pg 7 line 154: As a matter of facts(sic)
  3. Pg 11, line 172: ...immediate or two-times (sic)...

Perhaps it should be phrased: 'immediate or two-stage reconstruction', or ...'reconstruction in two stages'...

  1. Pg 18, line 240: ...'basingon experience'...
  2. Pg 18, line 252: ...'it can be usefully be employed'...

Here again both sentence structures are not ideal.

Apart from the above minor grammatical issues, there is reference to 

  1. Figure 9. It might be good for the authors to explain why detecting vascularity in what is assumed to be a fibroadenoma is considered an improvement in imaging.  Conventionally, vascularity is a characteristic of indeterminate lesions which may warrant biopsy.  
  2. Figure 10.  Was the final diagnosis listed in the text or in the legend? It would be interesting to know the histology for this irregular lesion.

Thank you. 

REVIEWER 2

The manuscript shows a number of clinical scenarios where high-frequency transducers (20-30 MHz) can be usefully employed to scan the breast, and side-by-side images obtained with conventional breast frequencies and high frequencies are shown to compare. They concluded that high-frequency (20-30 MHz) probes are available for its optimal spatial resolution and anatomic detail.

This is an interesting topic. However, some issues need to be clarified:

  1. The optimal spatial resolution of high-frequency probes are an obvious advanteges because of its physical features, but its other excellence are not well demonstrated in this article. The specific cases and images presented in the article do not convincingly illustrate the superiority of high-frequency ultrasound, lack of scientificity and generality.
  2. It’s better to organize the manuscript follow the format of this journal. The figures are of different sizes and the layout is chaotic.
  3. There are several typos and grammar issues throughout the manuscript.

MANUSCRIPT REVISION

REVIEWER #1

  1. Pg 1, line 41: In the study from...(sic)

[rev1, comment1]

According to your suggestion, we made the requested change.

  1. Pg 3 Line 92: Patients becomesvery anxious for any change seen over theur breast (sic)...

[rev1, comment2]

Sorry. we corrected the error.

  1. Pg 7 line 154: As a matter of facts(sic)

[rev1, comment3]

According to your suggestion, we changed the text introducing the sentence with the word: “in fact”. We hope very much that you will find our change adequate.

  1. Pg 11, line 172: ...immediate or two-times (sic)...

[rev1, comment4]

According to your suggestion, we made the requested change.

  1. Pg 18, line 240: ...'basingon experience'...

[rev1, comment5]

According to your suggestion, we made the requested change.

  1. Pg 18, line 252: ...'it can be usefully be employed'...

[rev1, comment6]

According to your suggestion, we modified the text. Now the sentence is the following: “If a high-frequency (20-30 MHz) probe is available, it should be used in a targeted way to image the breast”.

We hope very much that you will find our change adequate.

  1. Figure 9. It might be good for the authors to explain why detecting vascularity in what is assumed to be a fibroadenoma is considered an improvement in imaging.  Conventionally, vascularity is a characteristic of indeterminate lesions which may warrant biopsy.  

[rev1, comment7]

A short paragraph was added to discuss the relevance of vascularization assessment

  1. Figure 10.  Was the final diagnosis listed in the text or in the legend? It would be interesting to know the histology for this irregular lesion.

[rev1, comment8]

Sorry, we forgot to enter the result of the histological examination, which we proceeded to insert in the corresponding figure legend.

Reviewer 2 Report

The manuscript shows a number of clinical scenarios where high-frequency transducers (20-30 MHz) can be usefully employed to scan the breast, and side-by-side images obtained with conventional breast frequencies and high frequencies are shown to compare. They concluded that high-frequency (20-30 MHz) probes are available for its optimal spatial resolution and anatomic detail.  

This is an interesting topic. However, some issues need to be clarified:

1. The optimal spatial resolution of high-frequency probes are an obvious advanteges because of its physical features, but its other excellence are not well demonstrated in this article. The specific cases and images presented in the article do not convincingly illustrate the superiority of high-frequency ultrasound, lack of scientificity and generality.

2. Its better to organize the manuscript follow the format of this journal. The figures are of different sizes and the layout is chaotic.

3. There are several typos and grammar issues throughout the manuscript. 

Author Response

Dear Editor, dear Reviewers,

we are pleased that you are willing to reconsider our manuscript for publication, pending MAJOR revisions. After giving careful consideration to the points that the referees have made, we revised our article according to reviewer’ recommendations.

We hope very much that you and your colleagues will find our changes adequate.

REVIEWER 1

Dear authors,

Thank you for your work.

There are a few areas where grammar and sentence structure could be improved for clarity.

For example,

  1. Pg 1, line 41: In the study from...(sic)

May i suggest: In the study by...

  1. Pg 3 Line 92: Patients becomesvery anxious for any change seen over theur breast (sic)...
  2. Pg 7 line 154: As a matter of facts(sic)
  3. Pg 11, line 172: ...immediate or two-times (sic)...

Perhaps it should be phrased: 'immediate or two-stage reconstruction', or ...'reconstruction in two stages'...

  1. Pg 18, line 240: ...'basingon experience'...
  2. Pg 18, line 252: ...'it can be usefully be employed'...

Here again both sentence structures are not ideal.

Apart from the above minor grammatical issues, there is reference to 

  1. Figure 9. It might be good for the authors to explain why detecting vascularity in what is assumed to be a fibroadenoma is considered an improvement in imaging.  Conventionally, vascularity is a characteristic of indeterminate lesions which may warrant biopsy.  
  2. Figure 10.  Was the final diagnosis listed in the text or in the legend? It would be interesting to know the histology for this irregular lesion.

Thank you. 

REVIEWER 2 

The manuscript shows a number of clinical scenarios where high-frequency transducers (20-30 MHz) can be usefully employed to scan the breast, and side-by-side images obtained with conventional breast frequencies and high frequencies are shown to compare. They concluded that high-frequency (20-30 MHz) probes are available for its optimal spatial resolution and anatomic detail.

This is an interesting topic. However, some issues need to be clarified:

  1. The optimal spatial resolution of high-frequency probes are an obvious advanteges because of its physical features, but its other excellence are not well demonstrated in this article. The specific cases and images presented in the article do not convincingly illustrate the superiority of high-frequency ultrasound, lack of scientificity and generality.
  2. It’s better to organize the manuscript follow the format of this journal. The figures are of different sizes and the layout is chaotic.
  3. There are several typos and grammar issues throughout the manuscript.

MANUSCRIPT REVISION

REVIEWER #2

  1. The optimal spatial resolution of high-frequency probes are an obvious advanteges because of its physical features, but its other excellence are not well demonstrated in this article. The specific cases and images presented in the article do not convincingly illustrate the superiority of high-frequency ultrasound, lack of scientificity and generality.

[rev2, comment1]

We agree that, obviously, the use of higher frequencies always increase the spatial resolution and the image quality. The main limitation is the depth of the target. The interest of our manuscript is that we identified a wide number of circumstances where the target is not deep and consequently it is possible to scan it with a very-high frequency transducer. In our opinion the images included in this manuscript show significant differences between the two probes. We agree that the difference is not so marked to change, for example, the BI-RADS categorization but this is not in the purpose of this preliminary pictorial review  

  1. It’s better to organize the manuscript follow the format of this journal. The figures are of different sizes and the layout is chaotic.

[rev2, comment2]

Sorry, we adjusted the size of the figures and paginated them according to the journal’s recommendations. We hope very much that you will find our changes adequate.

  1. There are several typos and grammar issues throughout the manuscript. 

[rev2, comment3]

A native speaker review was carried out. Thanks for the note.
